# Brainstem Modulates Parkinsonism-Induced Orofacial Sensorimotor Dysfunctions

**DOI:** 10.3390/ijms241512270

**Published:** 2023-07-31

**Authors:** Glauce Crivelaro Nascimento, Gabrielle Jacob, Bruna Araujo Milan, Gabrielli Leal-Luiz, Bruno Lima Malzone, Airam Nicole Vivanco-Estela, Daniela Escobar-Espinal, Fernando José Dias, Elaine Del-Bel

**Affiliations:** 1Department of Oral and Basic Biology, School of Dentistry of Ribeirão Preto, University of São Paulo, São Paulo 14040-904, Brazil; glauce.nascimento@usp.br (G.C.N.); jacobgabrielle@usp.br (G.J.); brunamilan@usp.br (B.A.M.); gabrielli.luiz@usp.br (G.L.-L.); bruno.malzone@usp.br (B.L.M.); airamvivanco@gmail.com (A.N.V.-E.); danielamesc@usp.br (D.E.-E.); 2Department of Integral Dentistry, Oral Biology Research Centre (CIBO-UFRO), Dental School-Facultad de Odontología, Universidad de La Frontera, Temuco 4811230, Chile; 3Department of Neuroscience, School of Medicine of Ribeirão Preto, University of São Paulo, São Paulo 14040-900, Brazil; 4Department of Physiology, School of Medicine of Ribeirão Preto, University of São Paulo, São Paulo 14040-900, Brazil

**Keywords:** brain function, pain, neuroscience/neurobiology, neurotoxicity, muscle biology

## Abstract

Parkinson’s Disease (PD), treated with the dopamine precursor l-3,4-dihydroxyphenylalanine (L-DOPA), displays motor and non-motor orofacial manifestations. We investigated the pathophysiologic mechanisms of the lateral pterygoid muscles (LPMs) and the trigeminal system related to PD-induced orofacial manifestations. A PD rat model was produced by unilateral injection of 6-hydroxydopamine into the medial forebrain bundle. Abnormal involuntary movements (dyskinesia) and nociceptive responses were determined. We analyzed the immunodetection of Fos-B and microglia/astrocytes in trigeminal and facial nuclei and morphological markers in the LPMs. Hyperalgesia response was increased in hemiparkinsonian and dyskinetic rats. Hemiparkinsonism increased slow skeletal myosin fibers in the LPMs, while in the dyskinetic ones, these fibers decreased in the contralateral side of the lesion. Bilateral increased glycolytic metabolism and an inflammatory muscle profile were detected in dyskinetic rats. There was increased Fos-B expression in the spinal nucleus of lesioned rats and in the motor and facial nucleus in L-DOPA-induced dyskinetic rats in the contralateral side of the lesion. Glial cells were increased in the facial nucleus on the contralateral side of the lesion. Overall, spinal trigeminal nucleus activation may be associated with orofacial sensorial impairment in Parkinsonian rats, while a fatigue profile on LPMs is suggested in L-DOPA-induced dyskinesia when the motor and facial nucleus are activated.

## 1. Introduction

Basal ganglia structures and correlated dopaminergic mechanisms participate in the regulation of orofacial movements and sensitivity [1]. It is well recognized that dopamine (DA) receptor subtypes, particularly those of the D1 and D2 families, play a fundamental role in the control of orofacial movements [2]. In that way, it was evidenced that co-stimulation of both D1-like and D2-like receptors, exclusively in the ventrolateral part of the striatum, induced repetitive jaw movements in rats [2].

Furthermore, DA deficiency in the striatum due, for example, to the loss of neurons in the Substantia Nigra, results in bradykinesia, tremor, dysphagia, ineffective esophageal motility, difficulties with speech, swallowing and breathing, alteration in proprioception, tactile sensitivity, besides painful symptomatology. These orofacial manifestations are observed in patients with Parkinson’s disease (PD) [3]. On the other hand, long-term DA replacement therapy with l-3,4-dihydroxyphenylalanine (L-DOPA), the most effective treatment for PD motor symptoms [4], induces the appearance of dyskinesia characterized by abnormal involuntary movements, including oral motor deficit [5]. Indeed, orofacial symptomatology during L-DOPA-induced dyskinesia (LID) includes an increased duration of the chewing cycle and the movement velocities during mastication, facial muscle rigidity, increased tongue movement and orofacial pain [6]. A mesolimbic-pallidal circuitry hyperfunction is one of the acceptable mechanisms for orofacial dyskinesia appearance [5]. In fact, considering the presence of D1 receptors in areas innervated by mesolimbic fibers, it is reasonable to expect that compounds that inhibit D1 receptors may improve orofacial dyskinesia. Studies on rats have also shown that stimulating D1 and D2 receptors in the ventrolateral striatum (VLS) and nucleus accumbens (NAc) induces repetitive orofacial movements [5,7,8].

Besides dopaminergic transmission, other neurotransmitter systems may be involved in orofacial movement alterations in motor diseases such as PD. Regarding glutamatergic function and various glutamate receptor subtypes [9], it is noteworthy that among GluN2A–D receptors, the GluN2C subtype is specifically present in cerebellar granule cells and does not seem to be involved in overall motor activity [10,11]. Therefore, the elimination of GluN2A, B, and D receptors serves as a foundational step in investigating the role of glutamate receptor subtypes in orofacial movements.

Despite the negative impact of parkinsonism-induced alterations on orofacial physiology, very few studies have focused on characterizing these changes and studying neural mechanisms that modulate these responses [12,13,14].

For decades, unilateral 6-hydroxydopamine (6-OHDA) lesioned rats have been employed to investigate the pathogenesis and therapy of PD [15,16]. We took advantage of this model of PD to study muscle action segregation concerning right and left side control. L-DOPA-induced orofacial dyskinesia is well characterized by present oral lateralization deviations and tongue protrusion. These motor activities present a neural control from the nuclei of the cranial nerves (trigeminal motor nucleus and facial nucleus) located in the brainstem [17]. Interestingly, basal ganglia also influence the output to these lower brainstem areas to exert direct effects on oral motor control [1].

In line with this evidence, the purpose of this study is to investigate (i) orofacial nociceptive sensibility, (ii) morphological and metabolic lateral pterygoid muscle properties and (iii) the recruitment of trigeminal (motor and spinal) and facial nuclei in terms of neuronal activation and glial response, resulting from the striatal 6-OHDA lesion and LID in a rat model. Orofacial nociception was evaluated by a formalin test, and the lateral pterygoid muscles were analyzed regarding myosin fibers, succinate dehydrogenase activity and inflammatory cells (macrophage and neutrophil) by hematoxylin and eosin. Concerning neuronal activation, Fos-B was detected in the facial, motor, and spinal trigeminal nucleus. These same nuclei were evaluated regarding astrocyte and microglia density.

## 2. Results

### 2.1. 6-OHDA Nigrostriatal Lesion Characterization and AIMs Scores

Apomorphine-induced rotational behavior was used as an indicator of severe injury of the nigrostriatal system (Figure 1A). The chronic L-DOPA treatment regimens induced AIMs and sensitization of contralateral turning behavior in 6-OHDA-lesioned rats that increased over time (Figure 1B,C). No AIMs were observed in animals treated with the vehicle.

### 2.2. Orofacial Hyperalgesia Induced by 6-OHDA Lesion and LID

Previous results have shown orofacial allodynic and hyperalgesia responses in 6-OHDA lesioned rats [5]. Aiming to validate our model in this topic and investigate this response on dyskinetic rats, we performed the orofacial formalin test. A 6-OHDA lesion significantly increased the formalin-induced face rubbing nociceptive behavior on the contralateral side of the lesion in the second phase compared to non-lesioned rats. Hemiparkinsonian rats chronically treated with L-DOPA also presented an increased hyperalgesia behavior compared to the VEH+VEH, 6OH+VEH, and VEH+L-DOPA groups (Figure 1D).

### 2.3. Parkinson’s Disease and LID Impact on Lateral Pterygoid Muscle

Besides pain sensibility, people with Parkinson’s disease display some orofacial motor manifestations, and clinical studies have also evidenced motor impairments from the orofacial region on LID [12]. Our work investigated morphologic and histologic LPM features in hemiparkinsonian and L-DOPA-induced dyskinetic rats.

There is an increase in slow skeletal myosin in the LPM of 6-OHDA lesioned rats on the contralateral side of the lesion. The treatment with L-DOPA reduced these levels in the contralateral muscles (*p* < 0.05; Figure 2A,B,E) and induced a reduction of myosin fibers in the right muscle.

The LPM metabolism pattern (oxidative or glycolytic) was demonstrated by SDH activity (Figure 2C,D,F). The group of rats with LID showed increased glycolytic fibers of ipsi- and contralateral LPM to injury compared to lesioned rats treated with vehicle or non-lesioned ones (*p* < 0.05; Figure 2D). Moreover, the qualitative analysis of H.E. indicated an endomysium rich in inflammatory cells (macrophage and neutrophil; Figure 2G) induced by the LID model.

### 2.4. Is the Trigeminal System Mobilized Due to Orofacial Changes Induced by 6-OHDA Lesion and LID?

Spinal and motor trigeminal nuclei are recruited for the transmission of orofacial sensorial and muscular information [13]. The facial nucleus, in agreement, modulates the transduction of the facial expression motor information [14]. Brainstem structures have also been shown to be involved in PD [15]. Here, we investigated FosB, astrocyte (GFAP) and microglia (OX-42) expression on these three nuclei in the 6-OHDA-lesion and LID rat models (Figure 3 and Figure 4).

In the trigeminal motor and facial nucleus, the total level of Fos-B-ir was increased in 6-OHDA lesioned rats treated with L-DOPA compared to non-lesioned ones or ones treated with the vehicle (*p* < 0.05; Figure 3A–D). Additionally, Fos-B expression is increased on 6-OHDA lesioned rats treated with vehicle compared to non-lesioned ones or lesioned treated with L-DOPA in the oral spinal nucleus (*p* < 0.05; Figure 3E,F).

GFAP and OX-42 expression increased in the 6-OHDA lesioned rats treated with L-DOPA compared to non-lesioned ones or treated with vehicle (*p* < 0.05; Figure 4A–F). There were no differences in the oral spinal and motor nucleus (Figure 4G) between the groups.

## 3. Discussion

The pathophysiology process occurring during 6-OHDA and LID-induced orofacial manifestations is the major question in our study. Our data indicate that (i) hemiparkinsonian and L-DOPA-induced dyskinetic rats are hyperalgesic in the orofacial region contralateral to the lesion. Besides a sensorial evaluation from the orofacial area, (ii) the composition and metabolism of the lateral pterygoid muscle was impaired on the lesion and dyskinesia models. Concerning neuronal activation, (iii) the facial and motor trigeminal nucleus presented increased Fos-B in dyskinetic hemiparkinsonian rats, while in the spinal trigeminal nucleus, this increase is detected in hemiparkinsonian rats without dyskinesia. Finally, (iv) astrocyte and microglia density are increased in the facial nucleus of dyskinetic rats.

Nociceptive thresholds of 6-OHDA lesioned rats exposed to an orofacial acute inflammatory challenge are reduced in accordance with previous results [13,18]. Dopamine has been proposed to play a pivotal role in chronic pain [19]. Accordingly, dopamine agonists such as apomorphine, d-amphetamine, and cocaine also produce analgesia during the orofacial formalin test in rats [20].

Concerning LID, however, this is the first description of a hyperalgesic response in the oral region. Pain often fluctuates with Parkinsonian motor disability [21,22], and clinical trials have suggested that pain is more common in dyskinetic patients [23]. Sustained or intermittent muscle contractions induce pain symptomatology, and many substances released from a hypercontracted muscle (by ischemia and inflammation) increase the mechanical sensitivity of muscle nociceptors [1]. This hypersensitization is the best-established peripheral mechanism explaining pain in oral muscles during, for example, LID.

Moreover, the involvement of opioid transmission in striatal output pathways has been hypothesized for the induction of LID [24]. Opioid peptides appear overexpressed in the dopamine-depleted striatum, in animal models of LID and in postmortem brain tissue from PD patients treated with L-DOPA [25,26]. Considering the direct and indirect efferent connections from the striatum to brainstem structures involved in the descending pain modulation and that the opioid system also directly modulates orofacial nociceptive processes [27], the opioid system may have a role in orofacial hyperalgesia during dyskinesia.

Besides hypersensibility, people with Parkinson’s disease display oral motor manifestations [28]. Our data are consistent in showing increased slow skeletal myosin in the left lateral pterygoid muscles from lesioned rats and decreased myosin during LID on ipsi- and contralateral muscles in the lesion.

Muscle fibers can adapt to environmental alterations by changing fiber type according to their molecular properties, which are mainly related to the expression of MyHC isoforms [29]. Sarcomeric myosin is a complex hexameric structure and is composed of four light-chain (MyLC) and two heavy-chain (MyHC) molecules. The main difference between the MyHC isoforms is their time converting ATP into energy, which determines the speed of actin-myosin segregation [30]. This velocity increases successively, from fibers that contain solely MyHC-I to those containing MyHC-IIA, MyHC-IIX, and MyHC-IIB [31]. According to this, MyHC-I is found in indefatigable slow-type fibers, and MyHC-IIA, MyHC-IIX, and MyHC-IIB can be found in fast-type fibers.

Our data show reduced expression of MyHC-I LPM during LID, indicating less indefatigable fibers, which suggests a tendency for this muscle to become fatigued. The increased MyHC-1 on muscles in lesioned rats, in contrast, is indicative of more slow fibers, which reproduce the classic muscle hypofunction found in Parkinsonian patients [32].

In the same line, glycolytic fibers are increased on the contralateral muscle to the lesion from L-DOPA-induced dyskinetic rats. Succinate dehydrogenase (SDH) is a mitochondrial membrane protein integrant of the electron transport chain during cell respiration [33]. In our study, the muscle fibers were classified as oxidative or glycolytic, according to SDH detection [34]. The ability of the muscle to resist fatigue is proportional to the oxidative SDH muscle fiber amount. In this sense, a predominance of glycolytic metabolism is prejudicial for the muscle. In concern, a glycolytic metabolism associated with reduced MyHC-1 fibers strongly suggests a fatigued state on the studied muscle in LID. H.E. staining confirms this proposition since LID induced an endomysium rich in inflammatory cells, which, according to the literature, is one of the factors related to muscle fatigue [35].

Until now, our sensorial and motor evidence from the orofacial region appointed important stomatognathic system alterations after the Parkinsonian lesion and LID. Complementarily, the analysis of the neural substrates (trigeminal and facial complexes) participating in these modulations is originally described in this study. The Fos-B staining revealed a spinal (Sp5O) and motor (Mo5) trigeminal nucleus recruitment in 6-OHDA-lesioned and L-DOPA-induced dyskinetic rats, respectively. Moreover, the facial nucleus (7N) is activated during LID.

Immediate-early gene expression (such as ΔFosB) can be detected in structures outside the basal ganglia, and this finding is associated with LID [36]. Maegawa et al. [13] have found that the 6-OHDA-parkinsonism model increased c-Fos expression in the Sp5O after formalin was injected into the upper lip ipsilateral to the lesion.

The alteration found in Mo5 reflects the effect of LID in affecting all dopamine receptors in the striatum. In this perspective, GABAergic neurons arising from the ventrolateral part of the striatum primarily converge onto neurons in the dorsolateral part of the substantia nigra pars reticulate [37] and in the intern globus pallidum. From these areas, there is a projection of fibers to the reticular region around the Mo5 and parvicellular reticular formation of the medulla oblongata via the superior colliculus. In that region, many premotor neurons for the orofacial motor nuclei were known to be spread [38].

The lateral pterygoid muscle is connected to motor neurons in the dorsolateral portion of Mo5, so it is coherent with an alteration in neuronal activation in this region after changes in this muscle due to LID. Furthermore, besides the motor impairment per se, recent works have shown an increase in microglia and astrocytes in the Mo5 on trigeminal neuropathic pain models. In that way, the inhibition of microglial activity and attenuation of neuropathic pain behavior by microglial blockers suggest that microglial activity has a pathogenetic role also in orofacial motor dysfunction in neuropathic disease [39].

Further, injury to the facial and hypoglossal nerves also increases microglial activity in the motor nuclei of these nerves [40]. It is possible that similar to sensory nuclei, pro-inflammatory mediators, released by glial cells, modulate the excitability of motor neurons and thereby alter motor functions [39]. Considering that LID increases hyperalgesia in the present data, it is possible that the activation of glia in the facial nucleus may be explained not only by oral motor impairment but also by hypernociceptive behavior.

It is important to note that potential functional consequences of the molecular findings of this study, both sensory, muscular, and neural, can be observed in the orofacial region of Parkinson’s patients. The orofacial complex displays numerous signs of PD that are occasionally observed early in the disease process and, at that stage, typically respond to dopaminergic medication. However, they are more commonly observed later in the disease process when they become resistant to treatment. The usual blink rate, which is between 12 to 20 blinks per minute, is significantly reduced. Additionally, there is a limitation in upward gaze, and a masklike, impassive facial appearance develops due to reduced movements of the small facial muscles (hypomimia). Parkinsonian tremors are evident in the forehead, eyelids, lip and tongue muscles, as well as in involuntary movements of the jaw. Tremor and rigidity of the orofacial musculature can lead to orofacial pain, discomfort in the temporomandibular joint, cracked teeth, dental attrition, and difficulties in controlling and retaining dentures [41].

Adding to the complexity of this problem is the extended time it takes for Parkinson’s disease patients to eat due to difficulties in manipulating and transporting food to the mouth, slow chewing (bradykinesia), reduced tongue movement leading to a loss in the formation and propulsion of the food bolus to the back of the oral cavity, as well as difficulty in swallowing (dysphagia) caused by pharyngeal motor deficits. Approximately 75 percent of individuals with PD experience drooling of saliva from the corners of the mouth, often accompanied by angular cheilosis, skin irritation, and odor [42].

The present study is not without limitations. Future studies should examine potential sex differences in the orofacial alterations in Parkinsonism. Further, other masticatory and facial muscles could be explored. For future research directions, the exploration of the intricate circuitry of the basal ganglia involved in regulating motor behaviour, with a specific emphasis on orofacial movement, would be interesting. In this sense, by concentrating on techniques to quantify the topographical diversity of orofacial movements in both rats and genetically modified mice, we can gain deeper insights into the interactive and independent functions of the DAergic, GABAergic, and glutamatergic systems.

## 4. Materials and Methods

### 4.1. Subjects

Male Wistar rats (*n* = 96; Ribeirao Preto, Brazil; 200–250 g body weight) were housed under a 12 h light/dark cycle with free access to food and water, according to the guidelines for the care and use of mammals in US National Institutes of Health Guide for Care and Use of Laboratory Animals. The local Ethical Committee approved all experimental protocols (2016.1.784.58.0). Mounting evidence indicates that biological sex plays a significant role in the development and phenotypical expression of Parkinson’s disease (PD). The risk of developing PD is approximately twice as high in men compared to women, and this study used male rats for the analysis.

All procedures were performed during the light phase, in the morning period (8:00–12:00 a.m.) and in the same laboratory. Moreover, the same person performed the behavioral tests and received training one month before the experiments started.

### 4.2. Drugs

Apomorphine hydrochloride was administered subcutaneously (s.c.) at a dose of 0.5 mg/kg. L-DOPA (20 mg kg^−1^ orally by gavage; Hoffman-LaRoche, Sao Paulo, Brazil) plus the peripheral DOPA-decarboxylase-inhibitor benserazide–HCl (5 mg kg^−1^) was dissolved in water.

### 4.3. A Dopaminergic Lesion with the Neurotoxin 6-Hydroxydopamine

Figure 5 presents an experimental line describing the sequence of experiments.

The stereotaxic surgery was conducted as described before [15]. Desipramine (10 mg/kg, i.p.) in a volume of 2 mL/kg sterile saline was administered in the rats 30 min before the surgery. The animals were anesthetized with 2,2,2-tribromoethanol (250 mg kg^−1^, i.p.) and fixed into the stereotaxic apparatus (David Kopf, model USA, 9:57). Rats received one deposit (coordinates from bregma: AP = −4.3; LL = −1.6; DV = −8.3) of 2.0 μL 6-Hydroxydopamine (6-OHDA, Sigma-Aldrich, St. Louis, MO, USA) into the right medial forebrain bundle.

Motor asymmetry following 21 days of unilateral lesioning of the nigrostriatal pathway was assessed by apomorphine—(0.5 mg kg^−1^ in 0.9% NaCl, subcutaneous, Sigma)—induced rotational behavior analysis using automated rotameter bowls (Columbus Instruments International, Columbus, OH, USA). Total turns ipsilateral to the lesion were recorded over 45 min. Only animals showing an individual mean >2 full contralateral turns per min were used in the study [16].

### 4.4. L-DOPA-Induced Abnormal Involuntary Movements (AIMs)

Long-term use of L-DOPA in patients with Parkinson’s disease is frequently associated with the development of AIMs known as L-DOPA-induced dyskinesia (LID). A rat dyskinesia scale was used to monitor AIMs [17,43,44]. Rats underwent a 15-day chronic administration of L-DOPA (20 mg kg^−1^ with benserazide 5 mg kg^−1^, given orally by gavage) to establish a stable expression of dyskinesia. For the behavioral study, animals were chosen based on AIMs scores exceeding 10 and severity grading higher than 2 in at least one AIMs subtype during the L-DOPA effect [45,46]. The evaluation of AIMs subtypes, including axial, limb, and orolingual movements, was performed for a duration of 1 min every 20 min, totaling 180 min after the administration of L-DOPA. AIMs were assessed based on their severity and amplitude. The rodent AIMs rating scale consisted of four subtypes, encompassing movements affecting the head, neck, and trunk (axial AIMs), involuntary movements of both distal and proximal forelimbs (limb AIMs), and dyskinetic-like orolingual movements (orolingual AIMs) [16].

The four AIMs subtypes were scored according to the severity scale ranging from 0 to 4. The rats were allocated to the non-dyskinetic group when AIMs were absent or only occasional (score 0–1) and to the dyskinetic group when dyskinesia was frequent or continuous (score 2–4).

### 4.5. Nociceptive Analysis—Orofacial Formalin Test

To evaluate the tonic chemogenic pain response of rats in the orofacial region, we performed an orofacial formalin test according to the previous description [47]. First, the rats were adapted to a testing chamber for 20 min. The animals were removed from the box, and a volume of 50 µL of 2% formalin or 0.9% saline solution was injected subcutaneously into the orofacial region between the nose and the upper lip (vibrissa area). Immediately rats were returned to the testing chamber, and the time they spent rubbing their face was recorded. According to Grabow and Dougherty [48], the orofacial formalin test can be characterized by two phases.

### 4.6. Animal Euthanasia

For the immunohistochemistry technique, rats were deeply anesthetized with tribromoethanol and rapidly perfused transcardially with cold 0.9% saline solution containing heparin (200 μL of heparin 25,000 UI per liter of solution) and sodium nitrite (1 g/L solution). Animals were then immediately perfused with Somogyi-Takagi’s fixative solution with 4% paraformaldehyde (PFA; pH 7.4; Sigma-Aldrich, St. Louis, MO, USA). A 30% sucrose cryoprotection in 0.1 M phosphate buffer (PB) was performed on tissues for 48 h.

For histological techniques, rats were anesthetized with tribromoethanol and were euthanasied by decapitation.

### 4.7. Immunohistochemistry

The brainstem was evaluated at the level of the trigeminal motor (Bregma −8.80 until −9.80 mm), spinal nucleus (oral part; Bregma −10.30 until −11.60 mm) and facial nucleus (Bregma −10.30 until −11.60). The regions were located according to the coordinates of the Paxinos and Watson [49] rat brain atlas. Lateral pterygoid muscles free-floating or brainstem sections (25 μm) were submitted to antigen recovery (heating sections for 30 min in a water bath at 90 °C in a 0.1 M citrate buffer solution; pH 6.0). After rinsing in washing buffer (PBS 0.1 M + 0.15% Triton-X; pH 7.4), sections were incubated for 30 min with 2% hydrogen peroxide (diluted in PBS+triton) to remove endogenous peroxidase activity. Nonspecific binding sites were blocked in a solution containing 5% BSA washing buffer and 5% normal serum for 1 h. Next, sections were incubated with the primary antibodies. For muscles, the Monoclonal Anti Myosin antibody (1:4000; Sigma-Aldrich, St. Louis, MO, USA) produced in mice was used. The primary antibodies used on the brains were the following: Fos-B (rabbit anti-Fos-B; 1:4000; Santa Cruz Biotechnology, Santa Cruz, CA, USA), GFAP (intermediate filament glial fibrillary acid protein; rabbit anti-GFAP; 1:1000, Sigma-Aldrich) and OX-42 (rabbit anti-OX-42; 1:400; Waco laboratory Chemicals, Waco, TX, USA). After primary antibody incubation for 48 h, sections were successively washed and incubated in a secondary antibody solution for 90 min (goat anti-rabbit; 1:400). Sections were then incubated using the avidin-biotin immunoperoxidase method for 2 h (Vectastain ABC kit, Vector Lab, Burlingame, CA, USA), and immunoreactivity was revealed by the addition of chromogen diaminobenzidine (Sigma).

### 4.8. Metabolism Profile—Succinate Dehydrogenase Activity (SDH)

To evaluate metabolic activity in the muscles, a histochemical technique was performed to demonstrate SDH [50]. The slides with sections of 10 μm thickness were subjected to a reaction in a solution containing 0.2 M sodium succinate as a substrate and nitroblue tetrazolium to reveal the fiber type. Each cross-section of muscle was systematically captured in five images with an ×20 objective, with the aid of an Olympus DP72 camera coupled to an OLYMPUS BX61 microscope. The sample size was determined based on the stereological calculations disclosed by Mandarin-De-Lacerda et al. [51] to be meaningful quantification of light, intermediate and dark fibers. Therefore, each image was superimposed on an 80-point system that allowed the counting of points achieved randomly on each type of fiber—high, intermediate and low metabolic activity—using the Image J software 2023 (National Institutes of Health—NIH). The determination of the metabolic activity of the different fiber types followed the criteria established by Peter et al. [52]. The dark fibers, the ones with the most oxidative metabolic activity, determined the metabolic activity of this muscle. The same examiner was responsible for quantifying the test, which prevented varied tone interpretation of the fibers.

### 4.9. Morphometric Analysis

Serial muscle transverse sections of the 10 μm thick lateral pterygoid muscle were stained with H&E following the manufacturer’s standard protocol (Newcomer Supply, Middleton, WI, USA). Stained slides were qualitatively analyzed for the presence or absence of inflammation. All the images were scanned at an ×20 magnification using an Olympus BX51 inverted microscope (Olympus Scientific Solutions, Waltham, MA, USA) with a scale bar of 100 μm.

### 4.10. Statistical Analysis

The behavioral and molecular data passed the Kolmogorov–Smirnov test for normality. Therefore, these results were analysed by a parametric approach. All analyses were performed parametrically with a paired *t*-test and a two-way repeated measures analysis of variance (ANOVA), as indicated in the figure legends. A post-hoc Bonferroni test was adopted. A value of *p* ≤ 0.05 was considered statistically significant. The statistical program used was GraphPad Prism software 7.

## 5. Conclusions

Altogether, our results indicate, for the first time, that dyskinesia caused by chronic L-DOPA treatment in parkinsonian-lesioned rats produces orofacial hypernociceptive behavior, changes in masticatory muscle morphology and energetic metabolism. These changes seem to be modulated by a distinct pattern of neuronal and glial activation in essential brainstem nuclei participants from sensorial and motor orofacial responses, of which, Mo5 and 7N are the main structures responsible for the oral muscular changes.

## Figures and Tables

**Figure 1 ijms-24-12270-f001:**
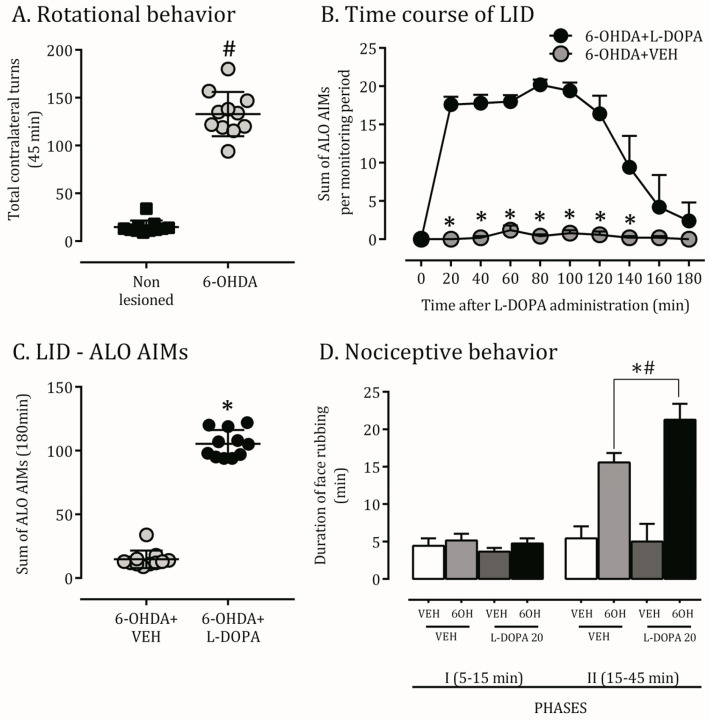
Behavioral and nociceptive alterations induced by 6-OHDA and L-DOPA chronic treatment. (**A**): The microinjection of 6-OHDA in the unilateral nigrostriatal pathway induced robust contralateral rotational behavior after apomorphine challenge; # *p* < 0.001, comparing 6-OHDA lesioned rats to no-lesioned ones. (**B**): Time course of development of abnormal involuntary movements (AIMs) by chronic administration of L-DOPA to 6-OHDA-lesioned; * *p* < 0.001, comparing lesioned rats treated with L-DOPA or its vehicle. (**C**): Sum of AIMs during 180 min; * *p* < 0.001, comparing lesioned rats treated with L-DOPA or its vehicle. (**D**): Nociceptive behavior detected by formalin test. The duration of face rubbing is quantified in two phases of the analysis (I and II); # *p* < 0.001, comparing lesioned rats (6-OHDA) with non-lesioned ones (VEH); * *p* < 0.001, comparing lesioned rats treated with L-DOPA (dyskinetic) or its vehicle. Data are mean ± SEM. Two-way ANOVA revealed a main effect of experimental condition (F(2,19) = 11.7), treatment (F(3,19) = 9.78) and an interaction (F(3,19) = 19.2; *p* < 0.05).

**Figure 2 ijms-24-12270-f002:**
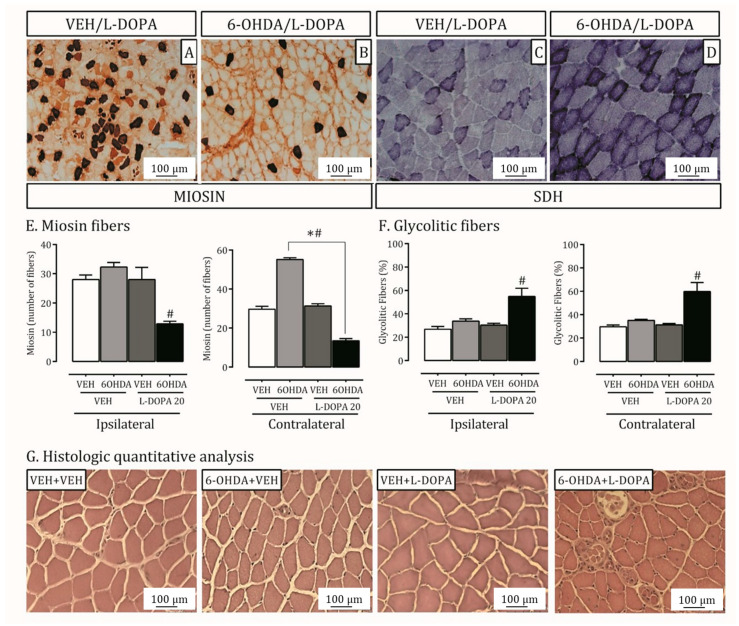
Injury in the lateral pterygoid muscle due to L-DOPA-induced dyskinesia. Lateral pterygoid muscles (LPM) from lesioned and non-lesioned rats treated with L-DOPA or its vehicle were processed immunohistochemically for Miosin (**A**,**B**) and histological SDH enzyme and inflammatory infiltrate (**C**,**D**,**G**) analysis. (**A**,**B**): Representative photomicrographs from LPM identifying Myosin fibers in non-lesioned (VEH/L-DOPA) and lesioned rats (6-OHDA/L-DOPA) treated with L-DOPA (20×). (**C**,**D**): Representative photomicrographs identifying glycolytic fibers through SDH histochemical reaction in non-lesioned (VEH/L-DOPA) and lesioned rats (6-OHDA/L-DOPA) treated with L-DOPA (40×). (**E**): Quantification of Myosin fibers on ipsilateral (**right**) and contralateral (**left**) LPM; # *p* < 0.001, comparing 6-OHDA lesioned rats to non-lesioned ones; * *p* < 0.001, comparing lesioned rats treated with L-DOPA (dyskinetic) or its vehicle. The two-way ANOVA revealed a main effect of treatment (ipsilateral; F(1,19) = 14.47; *p* < 0.05 contralateral; F(1,19) = 10.78; *p* = 0.001). (**F**): Quantification of glycolytic fibers on ipsilateral (**right**) and contralateral (**left**) LPM; # *p* < 0.05, comparing lesioned rats treated with L-DOPA (dyskinetic) with all other experimental groups. Data are mean ± SEM. The two-way ANOVA revealed a main effect of treatment (contralateral; F(1,19) = 11.88). (**G**): Representative photomicrographs from LPM staining with hematoxylin and eosin in four groups used in this study -lesioned (6-OHDA+VEH) and non-lesioned (VEH+VEH) treated with L-DOPA (6-OHDA+L-DOPA) or its vehicle (6-OHDA+VEH). Bar represents 100 μm. Non-lesioned or lesioned groups treated with vehicle presented muscular fibers with polygonal aspect and varied size, with peripheral nuclei. Within each fascicle, the endomysium has uniform spacing with the presence of capillaries and cells in the conjunctiva. Isolated points of endomysial thickening were evidenced near the surface of an irregularly shaped muscle fiber. In the experimental group with LID, however, was observed the endomysium with inflammatory cells (macrophage and neutrophil).

**Figure 3 ijms-24-12270-f003:**
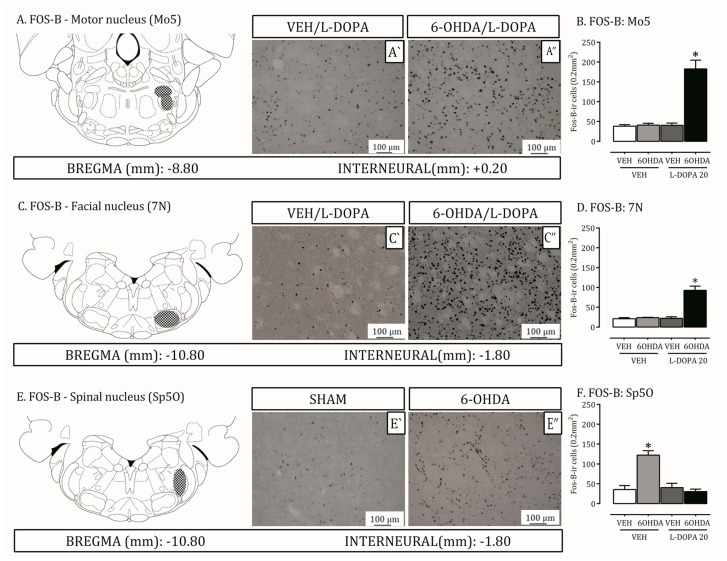
Activation of the Trigeminal system by L-DOPA in 6-OHDA-lesioned rats with dyskinesia. Two trigeminal nucleus (Motor: −8.80 to −9.80 and Spinal: −10.30 to −11.60) and the Facial nucleus (−10.30 to −11.60) were analyzed by Fos-B immunohistochemistry to the evaluation of their neuronal activation in non-lesioned (VEH/L-DOPA), and lesioned rats (6-OHDA/L-DOPA) treated with L-DOPA. Schematic representation of (**A**) motor nucleus (Mo5 bregma: −8.80), (**C**) facial nucleus (7N, bregma: −10.80) and (**E**) spinal nucleus (Sp5O, bregma: −10.80) from Paxinos and Watson (2006). Photomicrographs showing Fos-B-is in Mo5 (**A’**,**A”**); 7N (**C’**,**C”**) and Sp5O (**E’**,**E”**) (20×). Graphics showing the number of Fos-B-in cells induced by L-DOPA treatment in lesioned rats Mo5 (**B**); 7N (**D**) and Sp5O (**F**) * *p* < 0.05, compared with all other experimental groups. The two-way ANOVA revealed a main effect of treatment (ipsilateral; F(1,19) = 13.66).

**Figure 4 ijms-24-12270-f004:**
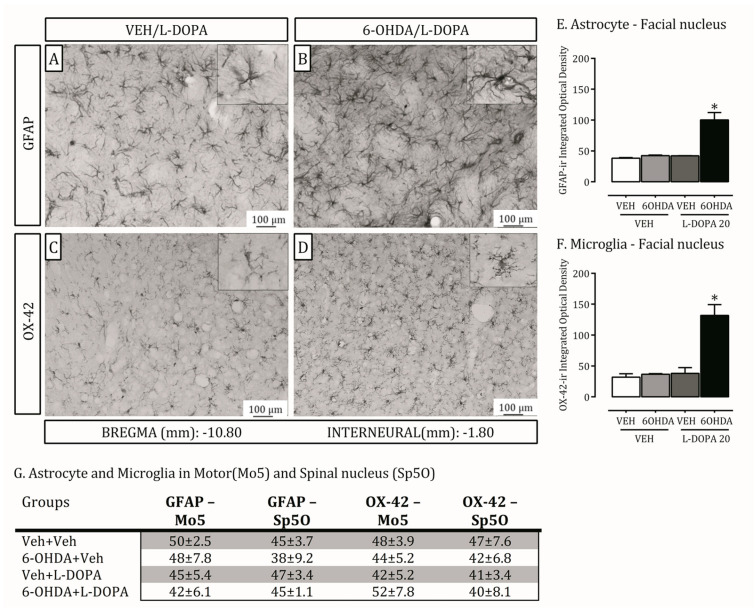
Neuroplasticity in the Facial nucleus through glial hyperactivity induced by L-DOPA-induced dyskinesia. Astrocyte (GFAP) and Microglia (OX-42) immunohistochemical in Motor (Mo5), Spinal (Sp5O) and Facial (7N) nucleus. (**A**–**D**): Representative photomicrographs of GFAP and OX-42 staining in 7N (Bregma −10.80) in non-lesioned (VEH/L-DOPA) and lesioned rats (6-OHDA/L-DOPA) treated with L-DOPA (20×). (**E**): Quantification of GFAP (optic density) in 7N; * *p* < 0.05, comparing lesioned rats treated with L-DOPA (dyskinetic) with all other experimental groups. (**F**): Quantification of OX-42 (optic density) in 7N; * *p* < 0.05, comparing lesioned rats treated with L-DOPA (dyskinetic) with all other experimental groups. (**G**): Quantification of GFAP and OX-42 (optic density) in Mo5 and Sp5O; there was no statistical difference between the groups. Data are mean ± SEM. Bar represents 100 um. The two-way ANOVA revealed a main effect of treatment (ipsilateral; F(1,19) = 9.64; *p* = 0.04).

**Figure 5 ijms-24-12270-f005:**
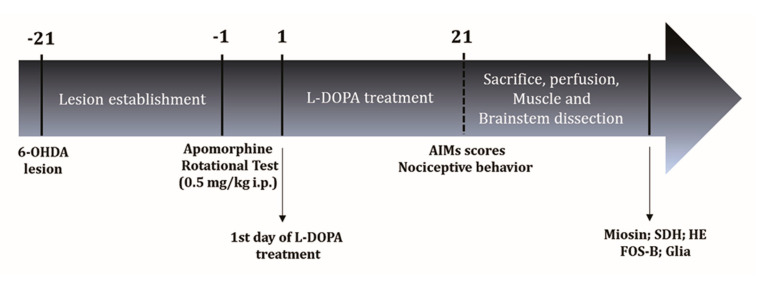
Experimental Design: Three weeks after the 6-OHDA lesion surgery, 48 animals were exposed to the apomorphine rotational test. Twenty-four of them constituted lesioned rats, and the other 24 were characterized as non-lesioned ones. Subsequently, 12 rats from each of the two groups were randomly assigned to be exposed to the drug treatment and the 12 remaining were chronically submitted to L-DOPA treatment. L-DOPA treatment started two days after the apomorphine rotational test. After chronic treatment with L-DOPA, rats were divided into experimental groups based on their dyskinesia score (AIMs scores). Moreover, on this last day, the experimental groups were submitted to a nociceptive behavior test, the animals were sacrificed, and lateral pterygoid muscle and brainstem were dissected for analyses (Miosin, SDH, HE, Fos-B and Glia). The nociceptive test and euthanasia were performed three hours after the last L-DOPA or vehicle administration. This experimental procedure was performed for immunohistochemistry analysis (*n* = 48 perfused animals) and muscle techniques (*n* = 48 non-perfused animals).

## Data Availability

The data that support the findings of this study are available from the corresponding author upon reasonable request.

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
