# Peer review of "Brainstem Modulates Parkinsonism-Induced Orofacial Sensorimotor Dysfunctions"

_ijms, 2023, doi:10.3390/ijms241512270_

Round 1
Reviewer 1 Report
Thank you very much for giving me the opportunity to review this very interesting manuscript entitled “Brainstem modulates Parkinsonism-induced oro-facial sensorimotor dysfunctions”. With some improvements the manuscript has the potential to be informative and comprehensive in the field of induced oro-facial sensorimotor dysfunctions due to Parkinsonism. The manuscript is generally well-written and organised. The style of writing is appropriate, although some sentences could be rephrased for improved flow and readability.
Introduction
The introduction clearly introduces the topic and the unique characteristics of this medical condition. However, it would benefit from a clearer statement of the objectives or the main points the authors intend to address. This would provide readers with a roadmap of what to expect throughout the manuscript.
It would be beneficial to expand on certain aspects. For instance, the discussion on the mechanisms that lead to the induced oro-facial sensorimotor dysfunctions due to Parkinsonism.
Methods
In Methods the manuscript provides a detailed description about the Methods. Maybe more details to be added about the animal model used, including the timing and route of L-DOPA administration.
Results
The authors presenting the data in an organised manner by using tables/figures to enhance clarity and facilitate the interpretation of the results. The tables are clear and appropriate.
Discussion
It would benefit the readers and the experts to be elaborated by the authors on the potential functional consequences of this condition of induced orofacial sensorimotor dysfunctions due to Parkinsonism.
Future research directions,
The authors did not provide adequate suggestions. Please provide more information.
Limitations
No Discussion about the limitations of the study, for example such as the reliance on a rat model. Please add possible limitations.
Structure
Please have the methods numbered as second in the manuscript and then 3rd the results and 4th the discussion.
Thank you.
Minor english editing
Author Response
Manuscript ID: ijms-2483406
Reviewers' comments:
Reply to the Reviewer #1:
Reviewer #1: Thank you very much for giving me the opportunity to review this very interesting manuscript entitled “Brainstem modulates Parkinsonism-induced oro-facial sensorimotor dysfunctions”. With some improvements the manuscript has the potential to be informative and comprehensive in the field of induced oro-facial sensorimotor dysfunctions due to Parkinsonism. The manuscript is generally well-written and organised. The style of writing is appropriate, although some sentences could be rephrased for improved flow and readability.
Response: We appreciate your careful review of our manuscript and your suggestions.
Introduction
The introduction clearly introduces the topic and the unique characteristics of this medical condition. However, it would benefit from a clearer statement of the objectives or the main points the authors intend to address. This would provide readers with a roadmap of what to expect throughout the manuscript.
It would be beneficial to expand on certain aspects. For instance, the discussion on the mechanisms that lead to the induced oro-facial sensorimotor dysfunctions due to Parkinsonism.
Response: Thank you for your suggestions. We added a sentence in the introduction (Page 2, Lines 83-87) with the objectives of the work: “Orofacial nociception was evaluated by formalin test and lateral pterygoid muscles were analyzed regarding myosin fibers, succinate dehydrogenase activity and inflammatory cells (macrophage and neutrophil) by hematoxylin and eosin. Concerning neuronal ac-tivation, Fos-B was detected in the facial, motor, and spinal trigeminal nucleus. These same nuclei were evaluated regarding astrocyte and microglia density.”
We also added an additional discussion about possible mechanisms to lead the induced orofacial sensorimotor dysfunctions due to Parkinsonism (Page 2, Lines 55-67): “In fact, considering the presence of D1 receptors in areas innervated by mesolimbic fi-bers, it is reasonable to expect that compounds that inhibit D1 receptors may improve orofacial dyskinesia. Studies on rats have shown also that stimulating D1 and D2 re-ceptors in the ventrolateral striatum (VLS) and nucleus accumbens (NAc) induces re-petitive orofacial movements (Koshikawa et al., 1990, 2011; Lee et al., 2003).
Besides dopaminergic transmission, other neurotransmitter systems may be in-volved in orofacial movements alterations in motor diseases such as PD. Regarding glutamatergic function and various glutamate receptor subtypes (Collingridge et al., 2009), it is noteworthy that among GluN2A–D receptors, the GluN2C subtype is spe-cifically present in cerebellar granule cells and does not seem to be involved in overall motor activity (Monyer et al., 1994; Ebralidze et al., 1996).
Therefore, the elimination of GluN2A, B, and D receptors serves as a foundational step in investigating the role of glutamate receptor subtypes in orofacial movements.”.
Methods
In Methods the manuscript provides a detailed description about the Methods. Maybe more details to be added about the animal model used, including the timing and route of L-DOPA administration.
Response: We improved the methodological details about L-DOPA administration and the dyskinesia model in rats (Page 10, Lines 340-351): “Rats underwent a 15-day chronic administration of L-DOPA (20 mg kg−1 with bense-razide 5mg kg−1, given orally by gavage) to establish a stable expression of dyskinesia. For the behavioral study, animals were chosen based on their AIMs scores exceeding 10 and severity grading higher than 2 in at least one AIMs subtype during the L-DOPA effect (Winkler et al., 2002; Cenci et al., 2007; Padovan-Neto et al., 2009). The evaluation of AIMs subtypes, including axial, limb, and orolingual movements, was performed for a duration of 1 minute every 20 minutes, totaling 180 minutes after the administration of L-DOPA. AIMs were assessed based on their severity and amplitude. The rodent AIMs rating scale consisted of four subtypes, encompassing movements affecting the head, neck, and trunk (axial AIMs), involuntary movements of both distal and proximal forelimbs (limb AIMs), and dyskinetic-like orolingual movements (orolingual AIMs)”.
Results
The authors presenting the data in an organised manner by using tables/figures to enhance clarity and facilitate the interpretation of the results. The tables are clear and appropriate.
Response: Thank you for your attention.
Discussion
It would benefit the readers and the experts to be elaborated by the authors on the potential functional consequences of this condition of induced orofacial sensorimotor dysfunctions due to Parkinsonism.
Response: We followed your suggestion and added the following part in the discussion section (Page 9, Lines 287-306): “It is important to note that potential functional consequences of the molecular findings of this study, both sensory and muscular, as well as neural, can be observed in the orofacial region of Parkinson's patients. The orofacial complex displays numerous signs of PD that are occasionally observed early in the disease process and, at that stage, typically respond to dopaminergic medication. However, they are more commonly observed later in the disease process when they become resistant to treatment. The usual blink rate, which is between 12 to 20 blinks per minute, is significantly reduced. Additionally, there is a limitation in upward gaze, and a masklike, impassive facial appearance develops due to reduced movements of the small facial muscles (hypomimia). Parkinsonian tremors are evident in the forehead, eyelids, lip and tongue muscles, as well as in involuntary movements of the jaw. Tremor and rigidity of the orofacial musculature can lead to orofacial pain, discomfort in the temporomandibular joint, cracked teeth, dental attrition, and difficulties in controlling and retaining den-tures (Friedlander et al., 2009).
Adding to the complexity of this problem is the extended time it takes for Parkin-son's disease patients to eat due to difficulties in manipulating and transporting food to the mouth, slow chewing (bradykinesia), reduced tongue movement leading to a loss in the formation and propulsion of the food bolus to the back of the oral cavity, as well as difficulty in swallowing (dysphagia) caused by pharyngeal motor deficits. Approxi-mately 75 percent of individuals with PD experience drooling of saliva from the corners of the mouth, often accompanied by angular cheilosis, skin irritation, and odor (Jankovic, 2008).”
Future research directions,
The authors did not provide adequate suggestions. Please provide more information.
Response: We are grateful for your suggestion. We added in the revised version some future research directions (Page 9, Lines 309-314): “. For future research directions, the exploration of the intricate circuitry of the basal ganglia involved in regulating motor behaviour, with a specific emphasis on orofacial movement is interesting. In this sense, by concentrating on techniques to quantify the topographical diversity of orofacial movements in both rats and genetically modified mice, we can gain deeper insights into the interactive and independent functions of the DAergic, GA-BAergic, and glutamatergic systems”.
Limitations
No Discussion about the limitations of the study, for example such as the reliance on a rat model. Please add possible limitations.
Response: We added a paragraph of limitations of the study in the discussion section (Page 9, Lines 307-314): “The present study is not without limitations. Future studies should examine po-tential sex differences in the orofacial alterations in parkinsonism. Further, other masticatory and facial muscles could be explored. For future research directions, the exploration of the intricate circuitry of the basal ganglia involved in regulating motor behaviour, with a specific emphasis on orofacial movement is interesting. In this sense, by concentrating on techniques to quantify the topographical di-versity of orofacial movements in both rats and genetically modified mice, we can gain deeper insights into the interactive and independent functions of the DAergic, GA-BAergic, and glutamatergic systems.”
Structure
Please have the methods numbered as second in the manuscript and then 3rd the results and 4th the discussion.
Response: Thank you for the suggestion. We modified the order of section according to your suggestion
Manuscript ID: ijms-2483406
Reviewers' comments:
Reply to the Reviewer #1:
Reviewer #1: Thank you very much for giving me the opportunity to review this very interesting manuscript entitled “Brainstem modulates Parkinsonism-induced oro-facial sensorimotor dysfunctions”. With some improvements the manuscript has the potential to be informative and comprehensive in the field of induced oro-facial sensorimotor dysfunctions due to Parkinsonism. The manuscript is generally well-written and organised. The style of writing is appropriate, although some sentences could be rephrased for improved flow and readability.
Response: We appreciate your careful review of our manuscript and your suggestions.
Introduction
The introduction clearly introduces the topic and the unique characteristics of this medical condition. However, it would benefit from a clearer statement of the objectives or the main points the authors intend to address. This would provide readers with a roadmap of what to expect throughout the manuscript.
It would be beneficial to expand on certain aspects. For instance, the discussion on the mechanisms that lead to the induced oro-facial sensorimotor dysfunctions due to Parkinsonism.
Response: Thank you for your suggestions. We added a sentence in the introduction (Page 2, Lines 83-87) with the objectives of the work: “Orofacial nociception was evaluated by formalin test and lateral pterygoid muscles were analyzed regarding myosin fibers, succinate dehydrogenase activity and inflammatory cells (macrophage and neutrophil) by hematoxylin and eosin. Concerning neuronal ac-tivation, Fos-B was detected in the facial, motor, and spinal trigeminal nucleus. These same nuclei were evaluated regarding astrocyte and microglia density.”
We also added an additional discussion about possible mechanisms to lead the induced orofacial sensorimotor dysfunctions due to Parkinsonism (Page 2, Lines 55-67): “In fact, considering the presence of D1 receptors in areas innervated by mesolimbic fi-bers, it is reasonable to expect that compounds that inhibit D1 receptors may improve orofacial dyskinesia. Studies on rats have shown also that stimulating D1 and D2 re-ceptors in the ventrolateral striatum (VLS) and nucleus accumbens (NAc) induces re-petitive orofacial movements (Koshikawa et al., 1990, 2011; Lee et al., 2003).
Besides dopaminergic transmission, other neurotransmitter systems may be in-volved in orofacial movements alterations in motor diseases such as PD. Regarding glutamatergic function and various glutamate receptor subtypes (Collingridge et al., 2009), it is noteworthy that among GluN2A–D receptors, the GluN2C subtype is spe-cifically present in cerebellar granule cells and does not seem to be involved in overall motor activity (Monyer et al., 1994; Ebralidze et al., 1996).
Therefore, the elimination of GluN2A, B, and D receptors serves as a foundational step in investigating the role of glutamate receptor subtypes in orofacial movements.”.
Methods
In Methods the manuscript provides a detailed description about the Methods. Maybe more details to be added about the animal model used, including the timing and route of L-DOPA administration.
Response: We improved the methodological details about L-DOPA administration and the dyskinesia model in rats (Page 10, Lines 340-351): “Rats underwent a 15-day chronic administration of L-DOPA (20 mg kg−1 with bense-razide 5mg kg−1, given orally by gavage) to establish a stable expression of dyskinesia. For the behavioral study, animals were chosen based on their AIMs scores exceeding 10 and severity grading higher than 2 in at least one AIMs subtype during the L-DOPA effect (Winkler et al., 2002; Cenci et al., 2007; Padovan-Neto et al., 2009). The evaluation of AIMs subtypes, including axial, limb, and orolingual movements, was performed for a duration of 1 minute every 20 minutes, totaling 180 minutes after the administration of L-DOPA. AIMs were assessed based on their severity and amplitude. The rodent AIMs rating scale consisted of four subtypes, encompassing movements affecting the head, neck, and trunk (axial AIMs), involuntary movements of both distal and proximal forelimbs (limb AIMs), and dyskinetic-like orolingual movements (orolingual AIMs)”.
Results
The authors presenting the data in an organised manner by using tables/figures to enhance clarity and facilitate the interpretation of the results. The tables are clear and appropriate.
Response: Thank you for your attention.
Discussion
It would benefit the readers and the experts to be elaborated by the authors on the potential functional consequences of this condition of induced orofacial sensorimotor dysfunctions due to Parkinsonism.
Response: We followed your suggestion and added the following part in the discussion section (Page 9, Lines 287-306): “It is important to note that potential functional consequences of the molecular findings of this study, both sensory and muscular, as well as neural, can be observed in the orofacial region of Parkinson's patients. The orofacial complex displays numerous signs of PD that are occasionally observed early in the disease process and, at that stage, typically respond to dopaminergic medication. However, they are more commonly observed later in the disease process when they become resistant to treatment. The usual blink rate, which is between 12 to 20 blinks per minute, is significantly reduced. Additionally, there is a limitation in upward gaze, and a masklike, impassive facial appearance develops due to reduced movements of the small facial muscles (hypomimia). Parkinsonian tremors are evident in the forehead, eyelids, lip and tongue muscles, as well as in involuntary movements of the jaw. Tremor and rigidity of the orofacial musculature can lead to orofacial pain, discomfort in the temporomandibular joint, cracked teeth, dental attrition, and difficulties in controlling and retaining den-tures (Friedlander et al., 2009).
Adding to the complexity of this problem is the extended time it takes for Parkin-son's disease patients to eat due to difficulties in manipulating and transporting food to the mouth, slow chewing (bradykinesia), reduced tongue movement leading to a loss in the formation and propulsion of the food bolus to the back of the oral cavity, as well as difficulty in swallowing (dysphagia) caused by pharyngeal motor deficits. Approxi-mately 75 percent of individuals with PD experience drooling of saliva from the corners of the mouth, often accompanied by angular cheilosis, skin irritation, and odor (Jankovic, 2008).”
Future research directions,
The authors did not provide adequate suggestions. Please provide more information.
Response: We are grateful for your suggestion. We added in the revised version some future research directions (Page 9, Lines 309-314): “. For future research directions, the exploration of the intricate circuitry of the basal ganglia involved in regulating motor behaviour, with a specific emphasis on orofacial movement is interesting. In this sense, by concentrating on techniques to quantify the topographical diversity of orofacial movements in both rats and genetically modified mice, we can gain deeper insights into the interactive and independent functions of the DAergic, GA-BAergic, and glutamatergic systems”.
Limitations
No Discussion about the limitations of the study, for example such as the reliance on a rat model. Please add possible limitations.
Response: We added a paragraph of limitations of the study in the discussion section (Page 9, Lines 307-314): “The present study is not without limitations. Future studies should examine po-tential sex differences in the orofacial alterations in parkinsonism. Further, other masticatory and facial muscles could be explored. For future research directions, the exploration of the intricate circuitry of the basal ganglia involved in regulating motor behaviour, with a specific emphasis on orofacial movement is interesting. In this sense, by concentrating on techniques to quantify the topographical di-versity of orofacial movements in both rats and genetically modified mice, we can gain deeper insights into the interactive and independent functions of the DAergic, GA-BAergic, and glutamatergic systems.”
Structure
Please have the methods numbered as second in the manuscript and then 3rd the results and 4th the discussion.
Manuscript ID: ijms-2483406
Reviewers' comments:
Reply to the Reviewer #1:
Reviewer #1: Thank you very much for giving me the opportunity to review this very interesting manuscript entitled “Brainstem modulates Parkinsonism-induced oro-facial sensorimotor dysfunctions”. With some improvements the manuscript has the potential to be informative and comprehensive in the field of induced oro-facial sensorimotor dysfunctions due to Parkinsonism. The manuscript is generally well-written and organised. The style of writing is appropriate, although some sentences could be rephrased for improved flow and readability.
Response: We appreciate your careful review of our manuscript and your suggestions.
Introduction
The introduction clearly introduces the topic and the unique characteristics of this medical condition. However, it would benefit from a clearer statement of the objectives or the main points the authors intend to address. This would provide readers with a roadmap of what to expect throughout the manuscript.
It would be beneficial to expand on certain aspects. For instance, the discussion on the mechanisms that lead to the induced oro-facial sensorimotor dysfunctions due to Parkinsonism.
Response: Thank you for your suggestions. We added a sentence in the introduction (Page 2, Lines 83-87) with the objectives of the work: “Orofacial nociception was evaluated by formalin test and lateral pterygoid muscles were analyzed regarding myosin fibers, succinate dehydrogenase activity and inflammatory cells (macrophage and neutrophil) by hematoxylin and eosin. Concerning neuronal ac-tivation, Fos-B was detected in the facial, motor, and spinal trigeminal nucleus. These same nuclei were evaluated regarding astrocyte and microglia density.”
We also added an additional discussion about possible mechanisms to lead the induced orofacial sensorimotor dysfunctions due to Parkinsonism (Page 2, Lines 55-67): “In fact, considering the presence of D1 receptors in areas innervated by mesolimbic fi-bers, it is reasonable to expect that compounds that inhibit D1 receptors may improve orofacial dyskinesia. Studies on rats have shown also that stimulating D1 and D2 re-ceptors in the ventrolateral striatum (VLS) and nucleus accumbens (NAc) induces re-petitive orofacial movements (Koshikawa et al., 1990, 2011; Lee et al., 2003).
Besides dopaminergic transmission, other neurotransmitter systems may be in-volved in orofacial movements alterations in motor diseases such as PD. Regarding glutamatergic function and various glutamate receptor subtypes (Collingridge et al., 2009), it is noteworthy that among GluN2A–D receptors, the GluN2C subtype is spe-cifically present in cerebellar granule cells and does not seem to be involved in overall motor activity (Monyer et al., 1994; Ebralidze et al., 1996).
Therefore, the elimination of GluN2A, B, and D receptors serves as a foundational step in investigating the role of glutamate receptor subtypes in orofacial movements.”.
Methods
In Methods the manuscript provides a detailed description about the Methods. Maybe more details to be added about the animal model used, including the timing and route of L-DOPA administration.
Response: We improved the methodological details about L-DOPA administration and the dyskinesia model in rats (Page 10, Lines 340-351): “Rats underwent a 15-day chronic administration of L-DOPA (20 mg kg−1 with bense-razide 5mg kg−1, given orally by gavage) to establish a stable expression of dyskinesia. For the behavioral study, animals were chosen based on their AIMs scores exceeding 10 and severity grading higher than 2 in at least one AIMs subtype during the L-DOPA effect (Winkler et al., 2002; Cenci et al., 2007; Padovan-Neto et al., 2009). The evaluation of AIMs subtypes, including axial, limb, and orolingual movements, was performed for a duration of 1 minute every 20 minutes, totaling 180 minutes after the administration of L-DOPA. AIMs were assessed based on their severity and amplitude. The rodent AIMs rating scale consisted of four subtypes, encompassing movements affecting the head, neck, and trunk (axial AIMs), involuntary movements of both distal and proximal forelimbs (limb AIMs), and dyskinetic-like orolingual movements (orolingual AIMs)”.
Results
The authors presenting the data in an organised manner by using tables/figures to enhance clarity and facilitate the interpretation of the results. The tables are clear and appropriate.
Response: Thank you for your attention.
Discussion
It would benefit the readers and the experts to be elaborated by the authors on the potential functional consequences of this condition of induced orofacial sensorimotor dysfunctions due to Parkinsonism.
Response: We followed your suggestion and added the following part in the discussion section (Page 9, Lines 287-306): “It is important to note that potential functional consequences of the molecular findings of this study, both sensory and muscular, as well as neural, can be observed in the orofacial region of Parkinson's patients. The orofacial complex displays numerous signs of PD that are occasionally observed early in the disease process and, at that stage, typically respond to dopaminergic medication. However, they are more commonly observed later in the disease process when they become resistant to treatment. The usual blink rate, which is between 12 to 20 blinks per minute, is significantly reduced. Additionally, there is a limitation in upward gaze, and a masklike, impassive facial appearance develops due to reduced movements of the small facial muscles (hypomimia). Parkinsonian tremors are evident in the forehead, eyelids, lip and tongue muscles, as well as in involuntary movements of the jaw. Tremor and rigidity of the orofacial musculature can lead to orofacial pain, discomfort in the temporomandibular joint, cracked teeth, dental attrition, and difficulties in controlling and retaining den-tures (Friedlander et al., 2009).
Adding to the complexity of this problem is the extended time it takes for Parkin-son's disease patients to eat due to difficulties in manipulating and transporting food to the mouth, slow chewing (bradykinesia), reduced tongue movement leading to a loss in the formation and propulsion of the food bolus to the back of the oral cavity, as well as difficulty in swallowing (dysphagia) caused by pharyngeal motor deficits. Approxi-mately 75 percent of individuals with PD experience drooling of saliva from the corners of the mouth, often accompanied by angular cheilosis, skin irritation, and odor (Jankovic, 2008).”
Future research directions,
The authors did not provide adequate suggestions. Please provide more information.
Response: We are grateful for your suggestion. We added in the revised version some future research directions (Page 9, Lines 309-314): “. For future research directions, the exploration of the intricate circuitry of the basal ganglia involved in regulating motor behaviour, with a specific emphasis on orofacial movement is interesting. In this sense, by concentrating on techniques to quantify the topographical diversity of orofacial movements in both rats and genetically modified mice, we can gain deeper insights into the interactive and independent functions of the DAergic, GA-BAergic, and glutamatergic systems”.
Limitations
No Discussion about the limitations of the study, for example such as the reliance on a rat model. Please add possible limitations.
Response: We added a paragraph of limitations of the study in the discussion section (Page 9, Lines 307-314): “The present study is not without limitations. Future studies should examine po-tential sex differences in the orofacial alterations in parkinsonism. Further, other masticatory and facial muscles could be explored. For future research directions, the exploration of the intricate circuitry of the basal ganglia involved in regulating motor behaviour, with a specific emphasis on orofacial movement is interesting. In this sense, by concentrating on techniques to quantify the topographical di-versity of orofacial movements in both rats and genetically modified mice, we can gain deeper insights into the interactive and independent functions of the DAergic, GA-BAergic, and glutamatergic systems.”
Structure
Please have the methods numbered as second in the manuscript and then 3rd the results and 4th the discussion.
Response: Thank you for the suggestion. We modified the order of section according to your suggestion.
.

Reviewer 2 Report
The authors have used male animals. It is recommended to use both females and males, based on the ARRIVE guidelines. Please justify the choice and add it in the method section.
The parametric tests have been used. It is assumed that the data are normally distributed, but there is no mention of the test of normality and its result in the statistic section. Please add.
It is not clear how these findings can be translated to the next phase. The authors are encouraged to add the limitations and potentials of the findings for translational purposes.
Author Response
Manuscript ID: ijms-2483406
Reviewers' comments:
Reply to the Reviewer #2:
Response: We appreciate your careful review of our manuscript and your suggestions.
The authors have used male animals. It is recommended to use both females and males, based on the ARRIVE guidelines. Please justify the choice and add it in the method section.
Response: Thank you for your observation. Mounting evidence indicates that biological sex plays a significant role in the development and phenotypical expression of Parkinson's disease (PD). The risk of developing PD is approximately twice as high in men compared to women (Solla et al., 2012; Zirra et al., 2022). This prevalence is one of the reasons why there are more studies on males than females in the literature. Again, unhappily, when we started the study, we were unable to keep female animals in our vivarium. This vivarium was recently modified, and we currently have these standardized conditions. We currently have a project under development that studies aspects of L-DOPA-induced dyskinesia in females.
The parametric tests have been used. It is assumed that the data are normally distributed, but there is no mention of the test of normality and its result in the statistic section. Please add.
Response: Thank you for your attention. We added this information in the Methods section in the revised version: “The behavioural and molecular data passed in the Kolmogorov–Smirnov test for normality. Therefore, these results were analysed by a parametric approach”.
It is not clear how these findings can be translated to the next phase. The authors are encouraged to add the limitations and potentials of the findings for translational purposes.
Response: We added a paragraph of limitations of the study and some future research directions in the discussion section (Page 9, Lines 307-314): “The present study is not without limitations. Future studies should examine potential sex differences in the orofacial alterations in parkinsonism. Further, other masticatory and facial muscles could be explored. For future research directions, the exploration of the intricate circuitry of the basal ganglia involved in regulating motor behaviour, with a specific emphasis on orofacial movement is interesting. In this sense, by concentrating on techniques to quantify the topographical diversity of orofacial movements in both rats and genetically modified mice, we can gain deeper insights into the interactive and independent functions of the DAergic, GA-BAergic, and glutamatergic systems.”
References used for responses:
Solla P, Cannas A, Ibba FC, Loi F, Corona M, Orofino G, Marrosu MG, Marrosu F (2012) Gender
differences in motor and non-motor symptoms among Sardinian patients with Parkinson’s disease. J Neurol Sci 323, 33–39.
Zirra A, Rao SC, Bestwick J, Rajalingam R, Marras C, Blauwendraat C, Mata IF, Noyce AJ. Gender Differences in the Prevalence of Parkinson's Disease. Mov Disord Clin Pract. 2022 Nov 14;10(1):86-93. doi: 10.1002/mdc3.

Round 2
Reviewer 1 Report
Dear Authors,
I read again your work entitled “Brainstem modulates Parkinsonism-induced orofacial sensorimotor dysfunctions”. I congratulate you for addressing all my comments.
Please have a final check for English and some remaining syntax errors.
Thank you.
Minor editing to the new texts added.